# Highly Sensitive Plasmonic Sensor Based on a Dual-Side Polished Photonic Crystal Fiber for Component Content Sensing Applications

**DOI:** 10.3390/nano9111587

**Published:** 2019-11-08

**Authors:** Nan Chen, Min Chang, Xuedian Zhang, Jun Zhou, Xinglian Lu, Songlin Zhuang

**Affiliations:** 1Shanghai Key Laboratory of Modern Optical System, School of Optical-Electrical and Computer Engineering, University of Shanghai for Science and Technology, Shanghai 200093, China; cn15800968586@163.com (N.C.); changmin@usst.edu.cn (M.C.); jz361@outlook.com (J.Z.); 151360021@st.usst.edu.cn (X.L.); slzhuang@yahoo.com (S.Z.); 2Shanghai Institute of Intelligent Science and Technology, Tongji University, Shanghai 200092, China

**Keywords:** photonic crystal fiber, fiber design, sensor, surface plasmon resonance

## Abstract

A plasmonic sensor based on a dual-side polished photonic crystal fiber operating in a telecommunication wavelength range is proposed and investigated numerically by the finite element method (FEM). We study the effects of structural parameters on the sensor’s performance and analyze their tuning effects on loss spectra. As a result, two configurations are found when the analyte refractive index (RI) changes from 1.395 to 1.415. For configuration 1, an RI resolution of 9.39 × 10^−6^, an average wavelength sensitivity of 10,650 nm/RIU (the maximum wavelength sensitivity is 12,400 nm/RIU), an amplitude sensitivity of 252 RIU^−1^ and a linearity of 0.99692 are achieved. For configuration 2, the RI resolution, average wavelength sensitivity, amplitude sensitivity and linearity are 1.19 × 10^−5^, 8400 nm/RIU, 85 RIU^−1^ and 0.98246, respectively. The combination of both configurations can broaden the wavelength range for the sensing detection. Additionally, the sensor has a superior figure of merit (FOM) to a single-side polished design. The proposed sensor has a maximum wavelength sensitivity, amplitude sensitivity and RI resolution of the same order magnitude as that of existing sensors as well as higher linearity, which allows it to fulfill the requirements for modern sensing of being densely compact, amenable to integration, affordable and capable of remote sensing.

## 1. Introduction

The surface plasmon resonance (SPR) phenomenon [1] has been extensively investigated both experimentally and theoretically by scholars’ tremendous efforts in recent decades. In general, the free electrons on a metal surface respond collectively by oscillating in resonance with a light wave, and the resonant interaction can constitute surface plasmons (SPs). Then, SPs are highly trapped on the interface between the metal and dielectric, and thus the strong confinement will lead to an electric field enhancement [2]. Simultaneously, the SPR effect has the trait of being very sensitive to slight refractive index (RI) changes. Combining the advantages of field enhancement and RI sensitivity, SPR sensors have been developed in many areas such as environment monitoring, food safety, water testing, gas detection, bio-sensing, and so on [3].

In the early stages, bulky Kretschmann prism-based SPR sensors played a certain role due to their robustness. However, the configurations with many optical and mechanical moving components are cumbersome, expensive and difficult to fabricate, so they cannot fulfill the requirements of modern sensors to be densely compact, amenable to integration, affordable and capable of remote sensing. To achieve modern sensing requirements, photonic crystal fibers (PCFs) [4] have become one of the most prominent potential candidates. PCFs have been widely recognized in the optical transmission field since 1996. They introduce the principle that the photonic crystal can modulate the electromagnetic wave with the corresponding wavelength into the fiber; therefore, cladding micro-nano dimension air pores arranged in the form of photonic crystals can regulate light propagation in PCFs. The excellent feature of PCFs is their design flexibility, so dispersion, birefringence, nonlinearity, etc. can be tailored by different air pore arrangements [5]. These aspects make PCFs particularly eye-catching in many areas and result in a wide range of applications in gas-based nonlinear optics, atom and particle guidance, ultrahigh nonlinearities, rare-earth doped lasers and sensing fields. Nowadays, the mature PCF fabrication widely utilized is the stack-and-draw procedure, which has proved highly versatile, allowing complex lattices to be assembled from individual stackable units of the correct size, and thereby shape, solid, empty, or doped glass regions can easily be incorporated. Because of natural microstructure channels in PCFs, gas, solid or liquid functional materials can be infiltrated or selectively infiltrated into periodic air holes, which helps to implement the manipulation of light–matter interactions [6]. Besides, the processing of conventional optical fibers, such as side polishing, rotation, slot and coating technology, can also be applied to PCFs, which makes PCFs’ application range extremely extensive.

It turns out that the future of PCF sensors based on the SPR effect can be expected, especially since Lee et al. achieved selectively filling gold wire into a PCF’s individual holes via experiments successfully combined with finite element simulation, which shows that gold-filled PCFs can be used as in-fiber wavelength-dependent devices [7]. A lot of novel and high performance PCF SPR sensor designs have emerged based on experimental or numerical methods. According to the processing technology for the PCF, the sensors can be roughly divided into three categories: First, selectively infiltrating metal into the PCF cladding air hole. Shuai et al. filled the central core pore with gold and designed a liquid-core PCF based on plasmonic effect with a maximum negative RI sensitivity of −5500 nm/RIU in the sensing range of 1.50–1.53 [8]. Rifat et al. proposed a highly sensitive plasmonic sensing scheme with miniaturized PCF attributes, which yielded maximum sensitivities of 11,000 nm/RIU and 1420 RIU^−1^, maximum resolutions of 9.1 × 10^−6^ and 7 × 10^−6^ RIU for the wavelength and amplitude sensing schemes and a maximum figure of merit (FOM) of 407 [9]. Second, coating a metal film around a PCF. Rifat et al. proposed a simple, two-ring, hexagonal lattice PCF biosensor with an active plasmonic gold layer, and the sensor could provide a maximum wavelength sensitivity of 4000 nm/RIU and a maximum amplitude sensitivity of 320 RIU^−1^ with a resolution of 3.125 × 10^−5^ RIU for wavelength interrogation (WI) and amplitude interrogation (AI) [10]. Lu et al. proposed a large-mode-area polymer PCF made of polymethyl methacrylate with the cladding having only one layer of air holes. A nanoscale metal film and analyte can be deposited on the outer side of the fiber, and an intensity sensitivity of 8.3 × 10^−5^–9.4 × 10^−5^ RIU can be obtained in their sensor [11]. Islam et al. presented a novel PCF sensor based on the SPR effect in the region from visible to near-infrared (500–2000 nm) wavelength range for RI sensing. The sensor showed a maximum wavelength sensitivity of 58,000 nm/RIU for the x polarization and 62,000 nm/RIU for the y polarization for analyte RI ranging from 1.33 to 1.43, maximum sensitivities of 1415 RIU^−1^ and 1293 RIU^−1^ for the x and y polarizations, a maximum figure of merit (FOM) of 1140 and fine RI resolution of 1.6 × 10^−6^ [12]. Third, using a side polishing technique to make a flat plane and coating a metal film on the plane as a sensing layer. Santos et al. presented a sensing configuration of RI, based on SPR in a micro-structured D-type optical fiber with a thin gold layer [13]. The sensor showed an improvement in sensitivity of 10 × 10^3^ nm/RIU and an improvement in resolution of 9.8 × 10^−6^ RIU. Luan et al. presented a D-shaped hollow core microstructured optical fiber (MOF)-based SPR sensor, achieving a spectral sensitivity of 2900 nm/RIU, a maximum amplitude sensitivity of 120 RIU^−1^, and a maximum phase sensitivity of 50,300 deg/RIU/cm [14]. What is more, other sensors with distinctive shapes and excellent performance are also desirable, with designs such as the opening-up dual-core micro-structured optical fiber-based SPR sensor by Luan et al. [15] and the SPR sensor with two open-ring channels based on a PCF for mid-infrared detection by Liu et al. [16], which was similarly compelling.

In practical sensing applications such as water quality monitoring, bio-medicine and chemistry, devices are required to be very sensitive to RI changes in some unknown analyte, so a highly sensitive PCF SPR sensor is proposed and numerally investigated in this paper. The sensor has a unique structure with dual-side polishing, which can help produce more obvious signals for detection to ensure sensing performance. On both planes, gold film is applied as a plasma excitation material and silicon nitride (Si_3_N_4_) layers are plated on top of the gold layer. Our sensor can be applied in the analyte environment to detect small RI changes. The effects of structural parameters such as the pore diameters of the inner and outer layers, pore pitch, thickness of the gold layer and Si_3_N_4_ layer are investigated. Furthermore, wavelength interrogation and amplitude interrogation methods [17] are implemented, and the major performance of the sensor is simulated by the finite element method (FEM) [18] using COMSOL software. 

## 2. Geometry Design and Theoretical Modeling

A schematic diagram of the structure of the dual-side polished PCF sensor is depicted in Figure 1. Different colors characterize different materials. There are two different sizes of pores, d_1_ and d_2_, to characterize outer and inner pores. Pore pitch is represented by Λ. d_1_, d_2_ and Λ directly affect the light transmission in the fiber core. The symbols *t_Au_* and *t_Si3N4_* represent the gold layer and Si_3_N_4_ layer thicknesses, respectively. The RI of the analytes varies between 1.395 and 1.415, which is denoted by n. The air RI in the pores is set to 1. The polishing depth is denoted by h; here h is equal to 0.65 Λ.

Fluoride phosphate (FP) glasses have various speciality optical fiber applications, including mid-IR lasers, UV transmission, and radiation dosimetry, of which N-FK51A soft glass is one of the most suitable types of glass for optical fibers [19]. Therefore, we select N-FK51A as the background material; its Sellmeier equation [20] can be given by:(1)$$n_{FK51A}^{2}(\lambda )=1+\frac{A_{1}\lambda^{2}}{\lambda^{2}-B_{1}}+\frac{A_{2}\lambda^{2}}{\lambda^{2}-B_{2}}+\frac{A_{3}\lambda^{2}}{\lambda^{2}-B_{3}}$$
where *A*_1_ = 0.971247817, *A*_2_ = 0.219014, *A*_3_ = 0.9046517, *B*_1_ = 0.00472302 μm^2^, *B*_2_ = 0.01535756 μm^2^, *B*_3_ = 168.68133 μm^2^, and *λ* is free space wavelength. 

When it comes to the selection of exciting plasma materials, gold and silver are commonly used as plasma excitation materials due to high excitation efficiency. However, gold has higher chemical stability than silver, and thus gold is usually favored. The dielectric constant of gold film coated on the side-polished flat plane can be defined by the Drude–Lorentz model [21,22]:(2)$$\varepsilon (\omega )=\varepsilon_{1}+i\varepsilon_{2}=\varepsilon_{\infty }-\frac{\omega_{p}^{2}}{\omega (\omega +i\omega_{c})}$$
where *ε*_∞_ is the dielectric constant of gold at high frequency and its value is 9.75, the plasma frequency of gold is expressed as $\omega_{p}=1.36\times 10^{16}$, and $\omega_{c}=1.45\times 10^{14}$ is the scattering frequency of electrons.

There is a Si_3_N_4_ layer on the gold film. This design has two functions: first, Si_3_N_4_ is a high RI material that enhances the restriction of light and adjusts the penetration depth of the evanescent field to control the intensity of the resonance; second, Si_3_N_4_ has high chemical stability and the upper area of the Si_3_N_4_ layer is employed as the sensing area, so the gold layer can be protected. K. Luke et al. [23] obtained data through experiments, and made these data fit to Lorentzian and Gaussian oscillator models, then simultaneously fit over both spectral ranges to obtain the following Sellmeier equation for Si_3_N_4_: (3)$$n_{Si3N4}^{2}=1+\frac{C_{1}\lambda^{2}}{\lambda^{2}-D_{1}^{2}}+\frac{C_{2}\lambda^{2}}{\lambda^{2}-D_{2}^{2}}$$
where *C*_1_ = 3.0249, *C*_2_ = 40,314, *D*_1_ = 135.3406, and *D*_2_ = 1,239,842. This equation is valid over the wavelength range of 310–5504 nm, with *λ* in the unit of nanometers.

When light enters the PCF along the axial direction, different modes will distribute on the x-y plane, and modal analysis can be implemented. The modal loss [24] can be characterized by the following expression:(4)αloss=8.686×2πλ×Im(neff)×104(dB/cm)
where *α_loss_* represents modal loss, and Im(*n_eff_*) is the imaginary part of effective refractive index (RI) for the fundamental mode.

Additionally, in the numerical calculation, perfect electric conductor (PEC) and perfect magnetic conductor (PMC) artificial boundary conditions are employed around the boundaries of the calculation area. In pursuit of higher precision, the perfectly matched layer (PML) is used to absorb scattered light. The triangular sub-domain is utilized to discretize the whole computation area. As a result, the computational region is meshed into 14,446 elements and the number of degrees of freedom equals 101,299.

## 3. Simulation Results and Discussions

### 3.1. Dispersion Relationship

The electric field patterns of x-polarization, y-polarization fundamental mode and surface plasmon polariton (SPP) mode are displayed in Figure 2a–c. Figure 2d shows that after SPP is activated, light is transmitted into the core in two manners, one is through the middle (marked in blue arrows and called manner1) and the other is on both sides (marked in yellow arrows and called manner2), and thus two configurations will occur in the sensor, corresponding to the two cases in Figure 3. Figure 3 shows the dispersion relations between y-polarization fundamental mode and SPP mode. It is well known that according to the coupled-mode theory, the incident light can be split into vertical and horizontal direction components, and only the y-polarization mode can couple with the SPP mode. When the phase matching condition between the y-polarization fundamental mode and the SPP mode is satisfied, two modes will couple, and the transmission spectrum of the incident light appears as a resonance peak, then gradually exponentially decays. At the two resonance wavelengths, there are two resonance peaks generated; they are named peak1 (left) and peak2 (right), respectively. The Img(neff) of the graph is the imaginary part of effective RI for y-polarization fundamental mode, which is proportional to the modal loss. 

### 3.2. Structural Parameter Effects

Generally, the performance of the sensor is greatly affected by structural parameters. Figure 4 displays the dependence of loss spectra for the different structural parameters. When d_1_ takes the values of 0.65 μm, 0.70 μm, and 0.75 μm and other parameters remain fixed, for peak1, the intensity of the resonance peak decreases slightly and resonance wavelength red shift occurs with the increment of d_1_; for peak2, resonance wavelength red shift also occurs, but the energy of the resonance peak drops markedly. The reason for this phenomenon is that the real part of effective RI for the y-polarization fundamental mode shifts down in the process of increasing d_1_. Furthermore, the dimension of the mode transmission channel is significantly reduced, and thus the amplitude variation of manner2 is observably greater than that of manner1. In Figure 4b, d_2_ can modulate the fundamental core mode directly, so the red shift phenomenon due to the increment of d_2_ is much more obvious than that caused by d_1_. Since the transmission channel of the SPP mode does not change, the small change in the core region does not have much influence on the resonance intensity; the intensity of peak1 and peak2 is only slightly reduced when d_2_ increases. The pitch Λ is also referred to as the lattice constant, which can modulate light in the entire PCF, so small magnitude changes can cause significant transmission loss variation, as shown in Figure 4c. When Λ increases by 0.02, both peak1 and peak2 are red shifted; the intensity of peak1 rises slightly and that of peak2 drops noticeably. Besides, the comparison between dual-side coating and single-side coating for the sensor is also investigated when d_1_ = 0.70 μm, d_2_ = 0.45 μm, Λ = 1.60 μm, t_Au_ = 50 nm, t_Si3N4_ = 50 nm and n = 1.410. In Figure 4d, it is clearly presented that the energy intensity of the dual-side coated sensor is significantly higher than that of the single-side coated sensor, and the distance between the two resonance peaks is also obvious. According to the two characteristics, the dual-side coated sensor produces two higher resonance peaks, and thus loss signals can be detected easily. Moreover, because the two loss peaks are far apart and easy to distinguish, the resonant wavelength interrogation will not be difficult and complicated. The appropriate resonance peak can be selected for different sensing detection applications. If the two resonance peaks are utilized reasonably, the sensing bands can cover the 1310 to 1550 nm communication window. Therefore, the dual-side coating scheme has superior sensing performance.

For the sensor, the effects of the film layer thickness on performance are self-evident, especially for the gold film as a plasmonic material. Figure 5a displays loss spectra for different Si_3_N_4_ layer thicknesses. When d_1_ = 0.70 μm, d_2_ = 0.45 μm, Λ =1.60 μm, t_Au_ = 50nm and n = 1.410, with an increment in Si_3_N_4_ layer thickness of 5 nm, two resonance peaks appear, both red shift, and the red shift of peak1 is larger than that of peak2, accompanied by varying degrees of intensity reduction. The explanation for this phenomenon is that Si_3_N_4_ with high RI has a strong binding effect on light, and thus when SPP mode transfers energy to the core, some of the light is trapped in the Si_3_N_4_ surface, which slightly weakens the intensity of the evanescent field; therefore, the loss peak is reduced progressively. However, the amount of Si_3_N_4_ is not large, so its effect is greater on the tuning resonance wavelength and protection. Figure 5b shows loss spectra for different gold layer thicknesses when d_1_ = 0.70 μm, d_2_ = 0.45 μm, Λ = 1.60 μm, t_Si3N4_ = 50 nm and n = 1.410. It can be clearly found that t_Au_ is decisive for the distance between the two resonance peaks. As the gold film increases, the two resonance peaks get close to each other, accompanied by increasing energy. Here, we can boldly deduce that when the thickness of the gold layer reaches a certain value, a very large resonance peak will appear at a certain wavelength.

### 3.3. Sensor Performance

Next, several important performance indicators in relation to the sensor were investigated. Based on the analysis of Section 3.2, the values of some fundamental parameters are fixed as d_1_ = 0.70 μm, d_2_ = 0.45 μm, Λ = 1.60 μm, t_Si3N4_ = 50 nm, t_Au_ = 50 nm and n = 1.410. The WI and AI methods are employed in analysis. Using the WI method, the RI variations can be detected by measuring the phase matching wavelength shift. In this view, the wavelength sensitivity can be calculated as:(5)$$S(\lambda )=\frac{\Delta \lambda_{peak}}{\Delta n_{a}}(nm/RIU)$$
where $\Delta \lambda_{peak}$ represents the resonance wavelength displacement in relation to RI changes, and $\Delta n_{a}$ represents variations in two adjacent RI (here $\Delta n_{a}=0.005$). Figure 6a shows the wavelength sensitivity of the sensor. It can be seen that SPR sensing characteristics are very sensitive to small RI changes. For peak1, the maximum $\Delta \lambda_{peak}$ is 62 nm when $\Delta n_{a}$ changes from 1.405 to 1.410, the corresponding wavelength sensitivity is 12,400 nm/RIU, and the other corresponding sensitivities are 9400 nm/RIU, 9600 nm/RIU, and 11,200 nm/RIU, successively, so the average wavelength sensitivity for the sensor is 10,650 nm/RIU. For peak2, the corresponding sensitivities are 6200 nm/RIU, 7200 nm/RIU, 11,600 nm/RIU and 8600nm/RIU, so its average wavelength sensitivity is 8400 nm/RIU. We find that the resonance peaks gradually drop as the RI increases, which can be explained by the fact that the higher the RI of the external environment becomes, the stronger the ability to bind light is, and the coupling efficiency between SPP mode and y-polarization mode is reduced, thereby causing the smaller amplitude. 

The wavelength resolution (*R_m_*) is another important parameter for this sensor, which is defined as:(6)$$R_{m}=\frac{\Delta \lambda_{\min}}{S(\lambda )}=\frac{\Delta n_{a}\cdot \Delta \lambda_{\min}}{\Delta \lambda_{peak}}(RIU)$$

Usually, the value of *R_m_* is the RI resolution for the sensor. Assuming $\Delta \lambda_{\min}$ = 0.1 nm, consequently, two excellent average RI resolutions of 9.39 × 10^−6^ and 1.19 × 10^−5^ are available from 1.395 to 1.415 in RI. Figure 6b demonstrates the extremely high linearity between analyte RI and resonance wavelength, the adjusted R^2^ is selected as the determined coefficient, and the coefficients corresponding to peak1 and peak2 are 0.99692 and 0.98246, respectively. Besides wavelength sensitivity, amplitude sensitivity is also a means of assessing the quality of a sensor. This sensitivity is defined as:(7)$$S_{A}(\lambda )=\frac{1}{\alpha (\lambda ,n_{a})}\frac{\partial \alpha (\lambda ,n_{a})}{\partial n_{a}}(RIU^{-1})$$
where $\alpha (\lambda ,n_{a})$ denotes modal loss when RI is equal to *n_a_*, $\partial \alpha (\lambda ,n_{a})$ is the loss difference due to RI variation and $\partial n_{a}$ is RI variation. The effects of analyte RI variations and different gold layer thicknesses on the amplitude sensitivity are achieved in Figure 6c,d by the AI method. When n = 1.395, maximum amplitude sensitivities of 252 RIU^−1^ for peak1 and 85 RIU^−1^ for peak2 are achieved. From Figure 6d, it can be seen that the sensitivity peaks are close each other and sensitivity decreases with the increasing of the gold film thickness. Therefore, the proper reduction of the film layer can not only help distinguish the two peaks, but also greatly improve the amplitude sensitivity of the sensor. Furthermore, for peak1, with the increase in the thickness t_Au_, sensitivity decreases obviously, while for peak 2, the sensitivity increases slightly. Generally, the damping loss will be higher if the gold layer is thicker. At this point, sensitivity should be reduced. However, the gold thickness can regulate the peak location, which results in two peaks expanding to the sides with the increasing of the gold thickness.

Furthermore, *FOM* [25,26] was considered. A larger *FOM* means a sharper peak, leading to a better detection precision. *FOM* can be given by:(8)$$FOM=\frac{m}{FWHM}$$
where *m* is the slope of resonance peak position per RI unit, and *FWHM* is the full width at half the maximum of the resonance peak. The calculation results of *FOM* are depicted in Figure 7. It is clearly found that the *FOM* of the dual-side coating sensor is superior to that of the single-side coating sensor. Adopting configuration 1, a huge *FOM* of 332 can be obtained. In fact, combining with previous performance parameter discussions, the sensor performance with configuration 2 is also reasonable, so the sensor structure can be applied in different bands with different configurations, which can greatly expand the application range. 

For comparison, the performance of the proposed PCF-SPR sensor in relation to previous research was investigated. The comparison results are shown in Table 1. It can be seen that the proposed sensor’s maximum wavelength sensitivity reaches the order of tens of thousands, an order of magnitude equivalent to the best reported sensors. Also, the maximum amplitude sensitivity reaches the order of hundreds, and the resolution can reach the order of 10^−6^, which are comparable to the best reported sensors. It is worth noting that the linearity of 0.99692 is better than the previously reported sensors. Therefore, our designed sensor has performance advantages in terms of wavelength sensitivity, amplitude sensitivity, resolution and linearity.

## 4. Conclusions

A highly sensitive plasmonic sensor based on a dual-side polished PCF is described and analyzed. The dual-side polishing design can provide two interfaces for light–matter interaction, and thus the evanescent field can be enhanced further. Correspondingly, the energy generated by the SPR effect is much higher than that of single-side polishing, which helps to enhance sensing performance. Considering the tolerance of the structural dimensions, d_1_ = 0.70 μm, d_2_ = 0.45 μm, Λ = 1.60 μm, t_Au_ = 50 nm and t_Si3N4_ = 50 nm are selected for investigating the performance. According to different transmission mechanisms, there are two resonance peaks when the phase match condition is satisfied. For peak1, an average wavelength sensitivity of 10,650 nm/RIU (the maximum wavelength sensitivity is 12,400 nm/RIU), an RI resolution of 9.39 × 10^−6^, an amplitude sensitivity of 252 RIU^−1^ and a linearity of 0.99692 are achieved when the analyte RI changes from 1.395 to 1.415; for peak2, the average wavelength sensitivity, RI resolution, amplitude sensitivity and linearity are 8400 nm/RIU, 1.19 × 10^−5^, 85 RIU^−1^ and 0.98246, respectively. Combining peak1 and peak2, the operating band of the sensor is greatly widened. By comparison, it is obvious that the FOM of the dual-side design is superior to that of the single-side design. The maximum wavelength sensitivity, amplitude sensitivity and resolution of the proposed sensor can reach the same order of magnitude as that of previous sensors, as well as higher linearity. These results can provide strong support for this sensor in environmental engineering, food safety, chemistry and biomedicine.

## Figures and Tables

**Figure 1 nanomaterials-09-01587-f001:**
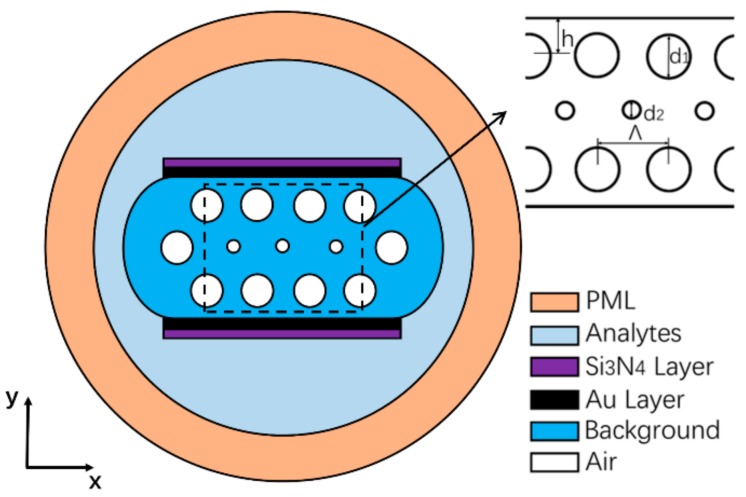
Schematic diagram of the proposed dual-side polished PCF sensor.

**Figure 2 nanomaterials-09-01587-f002:**
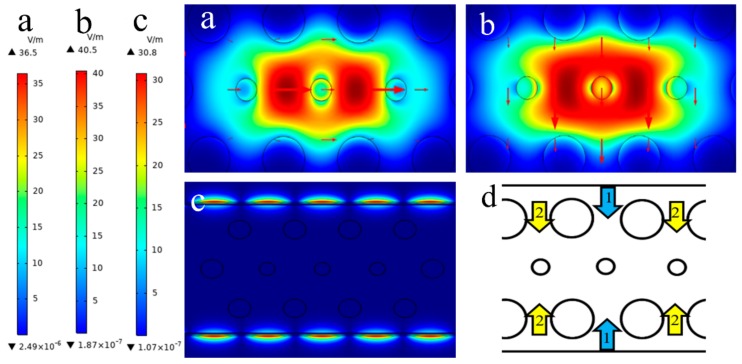
Electric field pattern: (**a**) x-polarization, (**b**) y-polarization fundamental mode, (**c**) surface plasmon polariton (SPP) mode above the Si_3_N_4_ layer. (**d**) Schematic diagram of two manners for SPP mode coupling into the fundamental core mode.

**Figure 3 nanomaterials-09-01587-f003:**
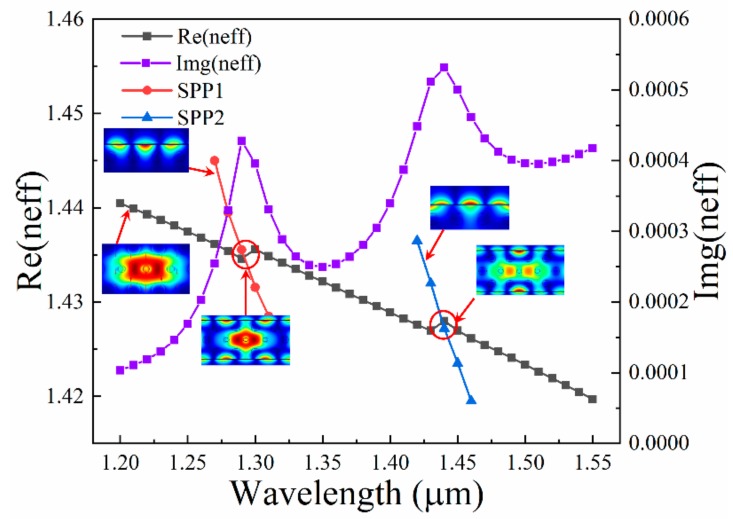
Dispersion relationship between y-polarization fundamental mode and SPP mode for the sensor. Illustrations of electric field distributions at key locations for phase matching are also displayed, when d_1_ = 0.70 μm, d_2_ = 0.45 μm, Λ = 1.60 μm, t_Au_ = 50 nm, t_Si3N4_ = 50 nm and n = 1.410.

**Figure 4 nanomaterials-09-01587-f004:**
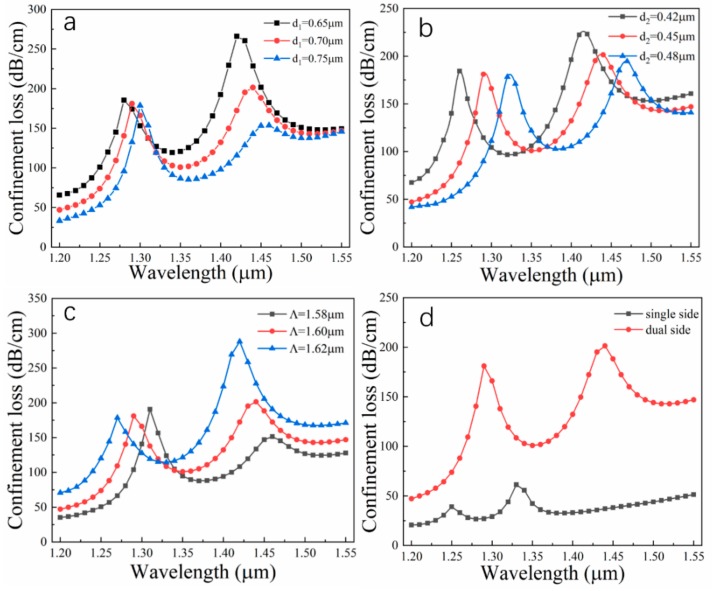
Dependence of loss spectra on different structural parameters: (**a**) different d_1_; (**b**) different d_2_; (**c**) different Λ; (**d**) comparison of dual-side coating and single-side coating. Here, t_Au_ = 50 nm, t_Si3N4_ = 50 nm and n = 1.410; the control variates method is adopted to analyze the performance.

**Figure 5 nanomaterials-09-01587-f005:**
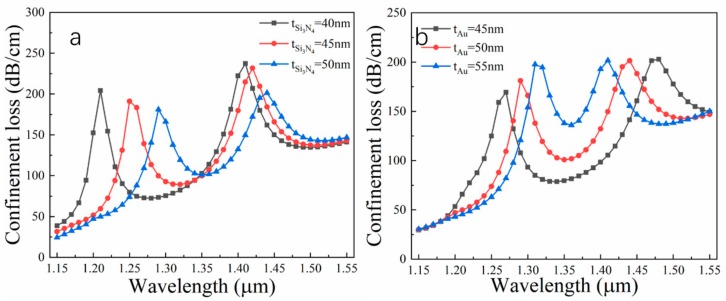
Dependence of loss spectra on different film thicknesses: (**a**) Si_3_N_4_ layer; (**b**) gold layer when d_1_ = 0.70 μm, d_2_ = 0.45 μm, Λ = 1.60 μm and n = 1410.

**Figure 6 nanomaterials-09-01587-f006:**
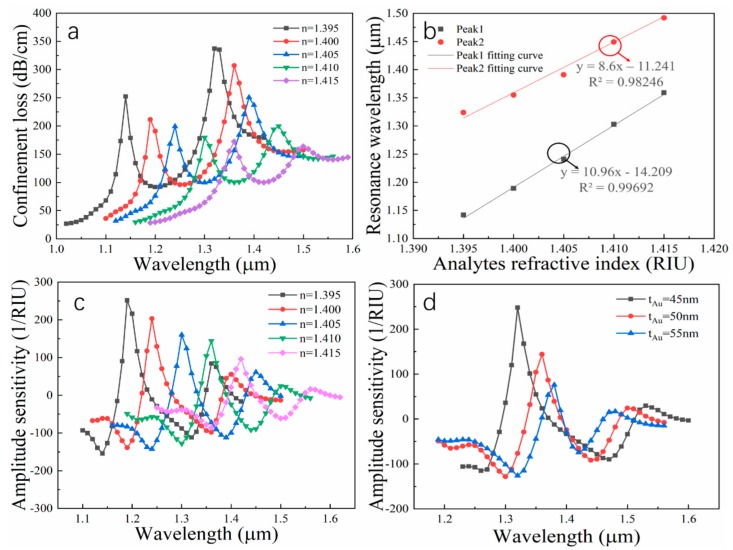
Performance parameters of the proposed sensor: (**a**) dependence of the loss spectra of the y-polarization fundamental mode on the analyte RI; (**b**) Linearity between the resonance wavelength and the RI of the analytes; (**c**) dependence of amplitude sensitivity on different analytes’ RI; (**d**) different gold thicknesses, n = 1.410.

**Figure 7 nanomaterials-09-01587-f007:**
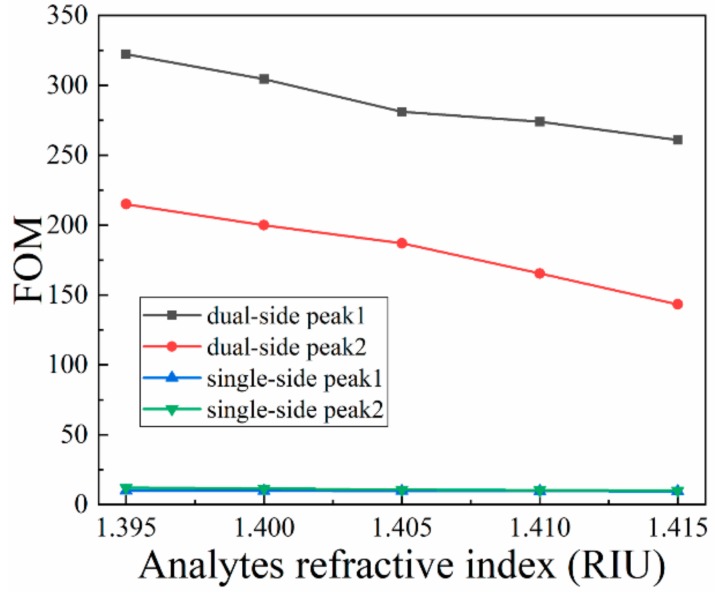
Dependence of figure of merit (FOM) of the two sensors on the analyte RI when d_1_ = 0.70 μm, d_2_ = 0.45 μm, Λ = 1.60 μm t_Au_ = 50 nm, t_Si3N4_ = 50 nm and n = 1.410.

**Table 1 nanomaterials-09-01587-t001:** Performance comparison of the proposed sensor and previous sensors.

Ref	Structure Type	Max. Wav. Sens. (nm/RIU)	Max. Amp. Sens. (RIU^−1^)	Resolution (RIU)	Linearity
[10]	Outer coating	4000	320	3.125 × 10^−5^	N/A
[12]	Dual-polarized	62,000	1415	1.6 × 10^−6^	N/A
[13]	Side polished	10,000	N/A	9.8 × 10^−6^	N/A
[16]	Open-ring channels	5500	333.8	7.69 × 10^−6^	N/A
[26]	Dual D-shape	43,200	1222	6.82 × 10^−6^	N/A
[27]	Side polished	23,000	820	1.22 × 10^−5^	0.9432
[28]	Dual-polarized	4600	420.4	N/A	0.9720
[29]	Multi-channel	4600	N/A	2 × 10^−5^	N/A
[30]	Side polished	5200	N/A	1.92 × 10^−5^	N/A
The present paper	Dual-side polished	12,400	252	9.39 × 10^−6^	0.99692

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
