# Peer review of "Highly Sensitive Plasmonic Sensor Based on a Dual-Side Polished Photonic Crystal Fiber for Component Content Sensing Applications"

_nanomaterials, 2019, doi:10.3390/nano9111587_

Round 1

Reviewer 1 Report

Although this manuscript might have potential to show actually interesting findings, it does not reach an appropriate standard to be acceptable for publication. The title is misleading is not too generic. What is the application authors intend to develop the sensor? The introduction does not provide the necessary background for understanding the study, I suggest to be more focused on the point.

Authors propose the device as a novel sensor, however, they should be more precise on what is novel and what is just a different architecture.

Manuscript is poorly formatted. It seems the result of a previous submission. Equations are reported as images, references are not correctly linked and formatted. Moreover, paragraphs should be divided according to MDPI template.

Manuscript should be revised. I found different errors (e.g. Based on the analysis of Section 3.2…it does not exist, lacks of unit is some cases).

L86 L87 why sensitivity and resolution have different unit?. Amplitude and wavelength sensitivity should distinguished

Going through the manuscript it seems it is a simulation study. Did authors perform any experimental studies?

Figure 2: A bar scale should be provided

Even though a table is reported for comparison no critical discussion was done. I suggest to critically compare the performance of the proposed sensor with the reviewed literature

Author Response

Reviewer #1

Comment 1: Although this manuscript might have potential to show actually interesting findings, it does not reach an appropriate standard to be acceptable for publication. The title is misleading is not too generic. What is the application authors intending to develop the sensor? The introduction does not provide the necessary background for understanding the study, I suggest to be more focused on the point. Authors propose the device as a novel sensor, however, they should be more precise on what is novel and what is just a different architecture.

Response: Thank you for your comments. Most of the PCF-SPR sensors are mainly used to detect some analytes quantitatively by the refractive index. Our paper also focuses this point. Thank you for your suggestions. Our initial title is generic indeed. In the revised manuscript, the title has been changed to “Highly sensitive plasmonic sensor based on dual-side polished photonic crystal fiber for component content sensing applications”. Also, necessary background has been added on the last paragraph (Line 1-Line 5) of Introduction section. About the novelty, we are sorry we didn’t express clearly. To make the novelty expression clearly, some explanations have been added on the conclusion. Actually, we want to express the point that our designed sensor possesses performance advantages including wavelength sensitivity, amplitude sensitivity, resolution and especially a higher linearity. About the architecture, the dual-side polishing design can provide two interfaces for light-matter interaction, and thus the evanescent field can be enhanced further. Correspondingly, the energy generated by the SPR effect is much higher than that of single-side polishing, which helps to enhance sensing performance.

Comment 2: Manuscript is poorly formatted. It seems the result of a previous submission. Equations are reported as images, references are not correctly linked and formatted. Moreover, paragraphs should be divided according to MDPI template.

Response: We are sorry that the manuscript is poorly formatted. The manuscript has been modified according to the MDPI template. The reference format and references have been modified. The formula is written using Math-type.

Comment 3: Manuscript should be revised. I found different errors (e.g. Based on the analysis of Section 3.2…it does not exist, lacks of unit is some cases).

Response: We are sorry for our mistakes. We have added the subtitle “3.2 Structural parameter effects” at the corresponding position on Page 5. And again we are sorry for our mistakes of lacks of unit. We examined carefully and found 3.2 lacks. Accordingly, we add the units. All the above mistakes have been corrected. Thank you for your comments.

Comment 4: L86-L87 why sensitivity and resolution have different unit? Amplitude and wavelength sensitivity should be distinguished.

Response: We are sorry we made the clerical mistakes in writing. The sensitivity and resolution units are modified on L14-L19 of the third paragraph in Introduction.

Comment 5: Going through the manuscript it seems it is a simulation study. Did authors perform any experimental studies?

Response: Thank you for your comments. As your comments, our work in this paper didn’t involve experimental research. We will try to perform the experimental studies in our future work. At present we put emphasis on the theoretical designs and performance analyses. According to the effective refractive index calculated by the finite element method, the sensing performance is calculated and analyzed.

Comment 6: Figure 2: A bar scale should be provided.

Response: Thank you for your guidance. We have added a scale bar in Figure 2.

Comment 7: Even though a table is reported for comparison no critical discussion was done. I suggest to critically compare the performance of the proposed sensor with the reviewed literature.

Response: Thank you for your guidance. We have added a new paragraph about some performance comparison and discussions as the last paragraph in Section 3.3.

Reviewer 2 Report

L-149, Check the English, “In pursuit of more accurate (some word(s) ?), …. In Ref. 16, the page number is lacking Judging from the performance comparison list of Table 1, your sensor showed almost similar performance to priori ones. I wonder if your claiming for having achieved great improvement over priori ones looks to be somewhat exaggeration. You are requested to reconsider the over-all tone in representation with respect to your contribution, especially in Abstract and Conclusion. Your spectroscopic response curves showed two distinguished resonances. I wonder if such not single- but dual (or complex)-resonance characteristics make the interrogation of resonance wavelength complicated and difficult as well as narrow the dynamic range of RI measurement, at the penalty of enhanced sensitivity. Some comments are requested.

Author Response

Reviewer #2

Comment 1: L-149, Check the English, “In pursuit of more accurate (some word(s)).

Response: We are sorry for our mistakes here. We have corrected the mistakes on the L3 of the last paragraph of Section 2. The correct expression should be “In pursuit of higher precision”. Thank you for your comments.

Comment 2: In Ref. 16, the page number is lacking.

Response: We are sorry for our mistakes here. The page numbers have been added. Besides, we found that a reference is missing and it should be the eleventh one. Accordingly, it has been added. ([11] Lu, Y.; Hao, C.; Wu, B.; Musideke, M.; Duan, L.; Wen, W.; Yao, J. Surface Plasmon Resonance Sensor Based on Polymer Photonic Crystal Fibers with Metal Nanolayers. SENSORS-BASEL. 2013, 13, 956-965.)

Comment 3: Judging from the performance comparison list of Table 1, your sensor showed almost similar performance to priori ones. I wonder if your claiming for having achieved great improvement over priori ones looks to be somewhat exaggeration. You are requested to reconsider the over-all tone in representation with respect to your contribution, especially in Abstract and Conclusion.

Response: We are sorry for our improper expressions and thank you so much for your comments. We checked carefully and revised the over-all tone expressions including Abstract and Conclusion. Revised portion are marked with red in the revised paper. And the detailed revisions are listed as following,

The word “Ultra-highly” on L1 of conclusion is revised as “highly”.

To make the paper more rigorous, we delete the word "novel" in the Abstract and explain the novelty in the Conclusion.

Comment 4: Your spectroscopic response curves showed two distinguished resonances. I wonder if such not single- but dual (or complex)-resonance characteristics make the interrogation of resonance wavelength complicated and difficult as well as narrow the dynamic range of RI measurement, at the penalty of enhanced sensitivity. Some comments are requested.

Response: We are sorry we didn’t express clearly. Actually, we want to show that the dual-side coated sensor produces two higher resonance peaks and thus loss signals can be detected easily. Moreover, because the two loss peaks are far apart and easy to be distinguished, the resonant wavelength interrogation will not be difficult and complicated. The appropriate resonance peak can be selected for different sensing detection applications. If the two resonance peaks are utilized reasonably, the sensing bands can cover the1310 and 1550 nm communication window. Therefore, the dual-side coating scheme has more excellent sensing performance. The corresponding explanations are added in the last 7 lines of the first paragraph in Section 3.2.

Round 2

Reviewer 2 Report

The corrections done are satisfactory to me.

This manuscript is a resubmission of an earlier submission. The following is a list of the peer review reports and author responses from that submission.